# Sweetness of Chilean Infants’ Diets: Methodology and Description

**DOI:** 10.3390/nu14071447

**Published:** 2022-03-30

**Authors:** Carmen Gloria González, Camila Corvalán, Marcela Reyes

**Affiliations:** Institute of Nutrition and Food Technology, University of Chile, Santiago 7830490, Chile; carmen.gonzalez@inta.uchile.cl (C.G.G.); ccorvalan@inta.uchile.cl (C.C.)

**Keywords:** diet, food, sweetness, infants

## Abstract

Sugars and other sweeteners contribute to the sweet taste of foods; exposure to this taste could alter appetite regulation and preferences for sweet products. Despite this, there is no widely accepted methodology for estimating overall diet sweetness. The objective of this study was to develop a methodology to estimate diet sweetness and describe diet sweetness in a cohort of Chilean infants. In order to estimate diet sweetness density, the sweetness intensity of foods was obtained from existing databases and from sensory evaluations in products with no available information and then linked to 24-h dietary recalls of infants at 12 and 36 months of age. Diet sweetness density was significantly and positively associated with total sugars and non-nutritive sweeteners intakes. The main food sources of sweetness at 12 months were fruits (27%) and beverages (19%). Sweetness density increased 40% between 12 and 36 months (from 1196 to 1673, *p* < 0.01), and sweetness density at both ages was significantly associated. At 36 months, beverages and dairy products were the main sources of sweetness (representing 32.2% and 28.6%, respectively). The methodology presented here to estimate the sweetness density of the diet could be useful for other studies to help elucidate different effects of exposure to high sweetness.

## 1. Introduction

High sugar consumption has been related to the development of various diseases, especially chronic non-communicable ones such as obesity, diabetes and cardiovascular diseases, among others [1]. For this reason, there are recommendations that aim to reduce free sugars intake to no more than 10% of energy consumed [1,2]. However, dietary information from different countries shows that sugars consumption is often high and above recommended levels [3,4,5,6]. As an alternative to sugars, non-nutritive sweeteners (NNS) have been incorporated by the food industry. These additives have been shown to be safe in terms of toxicity if consumed below the acceptable daily intake. Pediatric organizations have recognized there may be benefits of NNS for children, but also have expressed concerns and uncertainties about their use; several recommendations agree on avoiding their use among young children [7,8,9].

Sugars and NNS are characterized, among other things, by their sweet taste. There is an innate preference for sweet taste in the human species [10]. Consumption of sweet products can increase preferences for this taste, due to high palatability [11,12]. Likewise, it has been postulated that early exposure to sweet foods could influence a greater preference for these products in the future [11,13,14]; however, evidence is not conclusive [15]. Studies of this association are scarce and there is little information on the effects of diet sweetness, in both the short and long term.

Studies on diet sweetness have focused mainly on quantifying the consumption of specific foods and beverages, rather than on the general sweetness of the diet [16,17]. Estimates have also been made for dietary sugars content [18]. However, at present, a significant proportion of packaged foods contain NNS [19]. Thus, sugars content does not necessarily reflect the overall sweetness of the diet. In order to study the effects of diet sweetness, it is necessary to have methodologies that allow its complete quantification. However, methods are scant and thus, there are relatively few studies that have addressed this issue.

The sweetness intensity of foods and beverages can be evaluated using a descriptive profile—a sensory analysis tool that consists of describing sweetness or other organoleptic attributes, using unstructured scales, to characterize the intensity of each food. The description is made by a panel of previously trained experts [20]. We have identified three studies that have developed food sweetness databases [21,22,23], fundamental to estimate diet sweetness. In 2016, Lease et al. [21] published a database of the intensity of basic tastes (i.e., sweet, salty, sour, bitter and umami) for 377 foods. The work was carried out in Australia, by a trained sensory panel, with the aim of analyzing the relationship between tastes and food composition. Similarly, Martin et al. [22] published a database of basic tastes of 590 foods and preparations in 2014, from the results of a French trained sensory panel. Finally, van Langeveld et al. [23] developed a database of basic tastes of 469 foods, which was combined with information on food intake from 24-h recall (24R) surveys, to analyze the contribution of each group from tastes to energy and nutrient intake among children and adults in the Netherlands.

This article presents a methodology to estimate overall diet sweetness by assigning sweetness intensity values of foods and beverages, weighted by the amount consumed. The sweetness intensity value of foods (regardless of the type of sweetener used) was obtained from available databases and complemented with data obtained by sensory evaluation with a trained panel. Then, the overall dietary sweetness was applied to dietary data collected in children who participated in a longitudinal study at 12 and 36 months of age. We described diet sweetness, associations with different sociodemographic and dietary characteristics (i.e., intake of nutritive and NNS), and tracking of the indicator between both ages. We prioritized an infant and young children population to test this methodology because the first years of life are a critical period for food preferences, health and illness development [17,18,24]. Particularly, we chose 12 and 36 months of age because diets between those two ages are known to be significantly different; thus, it would allow us to test the sensitivity of our methodology for estimating sweetness intensity.

## 2. Materials and Methods

### 2.1. Design of Original and Current Study

We conducted secondary analyses of the Child Nutrition Study (ChiNut) cohort, conducted in Santiago de Chile, by the Institute of Nutrition and Food Technology (INTA), University of Chile. The ChiNut study was a double-blind randomized controlled trial (RCT), developed in collaboration between INTA, the Catholic University of Chile and the South East Metropolitan Health Service of Santiago, Chile. The objective was to evaluate the effect of a particular infant formula on the growth, health and brain development of children (*n* = 170) in the first two years of life, compared to regular infant formula (*n* = 170) and breastfeeding (*n* = 200) [25]. In the current study, we used dietary data collected at 12 and 36 months to conduct dietary sweetness estimates.

### 2.2. Sample

A total of 540 healthy infants, born in 2016 in Santiago, Chile were part of the original RCT. Inclusion criteria considered: singleton birth, up to 120 days of age at study registration, birth weight between 2500 to 4500 g, gestational age between 37 and 42 weeks, history of normal growth (weight between and inclusive of the 10th and 90th percentiles on the WHO growth chart); exclusion criteria were: use of complementary feeding, history of underlying conditions that could interfere with the ability of the infant to ingest food or to have normal growth and development, evidence of feeding difficulties or formula intolerance, immunodeficiency, and maternal illiteracy. The study was approved by the Ethics Committee at INTA’s IRB, University of Chile (N°19) on 13 December 2017, and the legal guardian of each participant provided informed consent [25]. For the current study, additional inclusion criteria included good health and complete dietary information from evaluations at 12 and 36 months. The final analytical sample for the current study was 436 children, exclusions were due to the lack of evaluation at 36 months (*n* = 104).

### 2.3. Anthropometry

Weight and length were measured in duplicate by trained nutritionists, using standardized protocols using a SECA scale and stadiometer, with a precision of 5 g (weight) and 0.1 cm (length), respectively. Body mass index (BMI) was computed and expressed as z-scores according to the World Health Organization (WHO) Child Growth Standards for sex and age; children were classified as normal weight (BMI z-score < +1SD), overweight (BMI z-score ≥ +1SD and <+2SD) and obesity (BMI z-score ≥ +2SD) [26]. The weight and height of the mothers of participants were also evaluated and BMI calculated. Mothers were categorized into normal weight, overweight and obesity, according to the WHO classification for adults [27].

### 2.4. Dietary Assessment

To assess participant dietary intake, mothers were surveyed by trained nutritionists, using a multi-step 24R [28], when children were 12 and 36 months of age. The SER-24H software (University of Chile; Santiago, Chile) developed by CIAPEC (Center for Research in Food Environments and Prevention of Nutrition-Associated Diseases) at INTA, University of Chile, was used for R24 assessment [29]. The software contains nutritional composition information of foods and beverages from the USDA Food Composition Database [30]; foods were selected by homologation, based on the nutritional information obtained from the Chilean nutritional composition database [31] and information available on nutritional labels [32,33]. To evaluate portions in homemade measurements, a photographic atlas of the National Survey of Food Consumption in Chile was used [34], which was complemented with photographs of measurements for specific products consumed by infants. Information on NNS content in packaged foods was obtained from manufacturer labels and was linked to the foods and beverages consumed by the study participants, proportional to the ingested portion [35]. In Chile, reporting of NNS content per portion of consumption and per 100 g or 100 mL of product is mandatory [36]. 

### 2.5. Sweetness Intensity Value in Food and Beverages

#### 2.5.1. Food Sweetness Intensity Databases

Two published food and beverages databases were used. The van Langeveld et al. database [23], with sweetness values between 1 and 100, and the Martin et al. database [22], with sweetness values between 1 and 10. These resources are publicly available and used similar methodologies (i.e., trained sensory panel). Foods included in SER-24H software were linked to the sweetness intensity value of these databases. Linkage was carried out by food groups, in addition to other characteristics (e.g., natural or processed foods and addition of sugars). We defined categories of foods based on the quality of the linkage. Equivalent foods were food in which the match was made with 100% certainty (e.g., natural products, such as bananas). Similar foods were foods with a linkage at the food group level, but that not necessarily were the same food. For example, different types of sweet cookies with chocolates were assigned the same sweetness intensity value reported for cookies with chocolates in the original database. Foods that were not matched corresponded to foods for which a similar food was not found, and therefore, a sweetness intensity value could not be assigned. The van Langeveld et al. database was the primary database used, as it was similar to the one used in the SER-24H software. The Martin et al. database was used for assigning sweetness intensity value to foods not matched (van Langeveld et al. database). After using both databases, there were 381 foods that were not matched (using both databases). As neither database reported breast milk sweetness intensity (both studies focused on foods consumed by adults), the value reported by McDaniel et al. was used [37]. For the remaining products (*n* = 380) a sensory evaluation was conducted. Additional details are presented in the results section.

#### 2.5.2. Sensory Evaluation of Sweetness

Products that were not assigned a sweetness intensity value from published databases were organized by food and beverage groups. From each group, a representative product (Appendix A Appendix A) was evaluated by a trained sensory panel (total *n* = 13).

All activities were conducted in the Sensorial Evaluation Laboratory at INTA, the University of Chile, which has physical space and individual cabins, designed according to the International Standard ISO 8589 [38]. Panelists were 12 healthy adults (25 to 55 years of age of both sexes) selected for the ability to recognize and discriminate tastes and to describe the sensory profile of products. Panel selection and training tests are presented in Appendix B. The training to describe the sweetness intensity profile of foods was carried out based on the procedures described in the International Standard ISO 8586 [39]. Panelists received a 6-h training for sweetness characterization of foods and beverages using a 10-point unstructured scale. Sucrose solutions and reference products described by van Langeveld et al. and also used by Martin et al. were utilized (Appendix A Appendix A). In the first part of the training, reference solutions were used, which contained different concentrations of sucrose in increasing amounts (2, 5, and 10%), representing specific values on the sweetness scale (13.33, 33.33, and 66.67%, respectively). Later, panelists were trained with commercial products used as references, with each product representing a specific value on the sweetness scale. To verify that the trained panel carried out the sweetness intensity evaluations in a similar way to the study by van Langeveld et al., foods that had a sweetness intensity value in the van Langeveld et al. database were incorporated into the sensory evaluation, to compare the intensities of sweetness assigned by both panels. The degree of agreement and reproducibility of the panelists was verified. Of the 23 trained panelists, the 12 best performers were selected for the final panel.

### 2.6. Estimation of Sweetness Density of Diet

An adaptation of the method described by Cox was used to estimate the sweetness density of the participant’s diet. [40]. Briefly, the sweetness density of the diet considered the sweetness intensity of each food or beverage consumed in a day, multiplied by the amount consumed, divided by the total amount of energy consumed in the day (in kcal), multiplied by 100. We estimated the mean sweetness density of diet consumed at 12 and 36 months of age for all participants.
Sweetness density = (∑ sweetness intensity level [UA] × portion consumed [g])/energy consumed per day [kcal]

The use of energy as the denominator could be misleading in diets with high content of foods/beverages using NNS, which might have no or low energy content. Thus, sensitivity analyses were conducted using an estimation of sweetness density using foods/beverages grams as the denominator instead of energy (Appendix C).

### 2.7. Statistical Analysis

Results were described as percentages for the categorical variables and mean and standard deviation for continuous variables. Variables that did not have a normal distribution were presented as median and interquartile range. Comparisons at the same age were made with the Mann–Whitney test or the Kruskal–Wallis test. Comparisons between different ages were made using the Student t-test for paired samples, in the case of continuous variables, and with the McNemar test, in the case of categorical variables. The independent association between sweetness density at 12 and 36 months was studied with linear regressions, considering covariates (participant age, participant and maternal weight status, and maternal education, both at 12 and 36 months). Associations between sweetness density and sweeteners intake were also studied with linear regressions. The main food sources of dietary sweetness were estimated as percent contributions of different food groups to the total sweetness density of diet consumed. Food groups are detailed in the Appendix A). The sample size was determined from the original RCT study. We set the statistical significance level at *p* < 0.05 and used Stata 15.0 software (StataCorp. 2017. Stata Statistical Software: Release 15. College Station, TX: StataCorp LLC, Texas City, TX, USA) for all analyses.

## 3. Results

### 3.1. Construction of the Food Sweetness Intensity Database

For a total of 5649 foods available in SER-24H, 5115 (91%) products were assigned sweetness intensity values from the van Langeveld et al. database, with an exact match of products (i.e., equivalent food) in 3140 (61.4%). From the remaining products (*n* = 534), 153 (2.7% of total foods) were matched using the Martin et al. database, with an exact match of products in 89 (58.2%). For those 153 products, sweetness intensity values were assigned based on the predicted values in the scale of the van Langeveld et al. (0–100 scale), derived from the association obtained from 4266 products that matched in both databases (beta coefficient for the regression of 11.05). After these subsequent matches, breastmilk sweetness intensity value was assigned from McDaniel et al. (0–100 scale); the remaining 380 foods were categorized into 13 food groups (Appendix A) and the sweetness intensity profile was evaluated for a representative food from each food group. This evaluation was carried out with a trained panel, which was comparable to that used to develop the primary sweetness database, according to the comparison of sweetness intensity values for 3 products (Table 1).

Table 2 shows examples of sweetness values (provided either by sugars or NNS) for some foods reported in the dietary recalls (either at 12 or 36 months). The average sweetness of fruits in their natural state was 32, that of commercial cola drinks (regular and diet) was 44 and that of natural pasta was 3.

### 3.2. Sweetness Density Estimation of Child Diet at 12 and 36 Months of Age

For this study, only the 1737 products that were consumed by the participants at 12 and 36 months of age were analyzed for estimating dietary sweetness. Of these, 1385 values (79.7%) were assigned from the van Langeveld et al. database, 14 values (0.8%) were predicted from the Martin et al. database, and 338 products (19.5%) had a value assigned by the trained panel. Sweetness intensity values for these 1737 foods are available in the Appendix A. Appendix A shows the results of sweetness intensity values used by the food group and the source of each value.

Table 3 presents the anthropometric and sociodemographic characterization of participants at 12 and 36 months of age. The sample was 49% female. At 12 months, 37% were overweight while at 36 months, this figure raised to 40% (*p* < 0.01). Most mothers had more than 12 years of education, at both measurements, with years of study significantly higher when children were 36 compared to 12 months (*p* < 0.01). More than 70% of mothers were overweight when children were 12 months of age, which significantly increased to over 80% at 36 months of age.

At 12 months, 19% of the sample received breastmilk the day before. The average intake of energy was 833 ± 296 kcal, from which 54 ± 9% same from carbohydrates and 29 ± 9% came from total sugars; 36% of participants consumed at least one NNS (either from processed foods or added at home). At 36 months, only 10% was breastfed; the average intake of energy was 1317 ± 608 kcal, 56 ± 7% from carbohydrates and 28 ± 8% from total sugars, and 76% consumed at least one NNS.

Sweetness density of diet was estimated at 12 and 36 months of life. At 12 months, we observed a median sweetness density of 1196 with an interquartile range of 817–1539. There were no differences by sex or categories of BMI (all *p*-values > 0.05). As expected, the sweetness density of the diet showed positive associations with total sugar intake expressed as percent of the energy consumed, as well as with NNS intake (Table 4). The main food sources of dietary sweetness density are presented in Table 5. At 12 months, fruits, beverages and vegetables were the most relevant groups contributing to total sweetness density, representing 64% of the total diet sweetness. These food sources of sweetness included both sugars and NNS. 

Sweetness density at 36 months was 1673 (1244–2260); males had higher values than females (*p* < 0.01), with no differences between categories of BMI (*p* > 0.05). Compared to 12 months, sweetness density was 40% higher at 36 months (*p* < 0.01). The sweetness densities of the diet at 12 and 36 months were significantly associated between them, both crude (β coefficients [95% confidence interval] = 0.22 [0.08; 0.36]) and adjusted by relevant covariates (0.26 [0.12; 0.40]), both *p*-values < 0.01. The increase in sweetness intensity between 12 and 36 months was significant for both sexes (an increase of 47%, from 1230 to 1806 among males and from 1147 to 1615 (40%) among females, both *p* < 0.05). Increases were also observed for all BMI categories: normal weight participants increased their sweetness density from 1188 to 1664 (40%), overweight participants increased from 1217 to 1779 (46%) and obese participants increased from 916 to 1633 (78%) (all *p*-values < 0.01). These analyses were repeated using sweetness density estimated with grams instead of energy as the denominator, obtaining similar results (Appendix C). 

Like at 12 months of age, the sweetness density of the diet at 36 months showed positive associations with total sugars intake and with NNS intake (Table 4). The food groups that contributed the most to diet sweetness were beverages, dairy products and fruits, representing 75% of diet sweetness at 36 months (Table 5). The proportions of food items consumed by our sample that contained NNS were 15% at 12 months of age and 14% at 36 months of age; the most common NNS was sucralose (10% of food items), acesulfame (5%), and aspartame (4%).

We were able to build a database with information on sweetness intensity for 5649 foods, based on previously published data and sensory evaluations. We used the developed database to evaluate the diet of a sample of young children, at 12 and 36 months. As expected, the sweetness density of the diet was associated with the intake of nutritive (i.e., % of energy derived from sugars) and non-nutritive sweeteners (i.e., consumption of at least 1 NNS); moreover, sweetness density at 12 months was associated with sweetness density at 36 months (which was 40% higher than at 12 months of age). At 12 months, the main food source of sweetness was fruits (27%), with beverages being the second source (19%); however, at 36 months, beverages were the primary source with 32%, followed by dairy products as the second source (29%). Fruit intake is advised by most health/nutrition professionals in primary care facilities as part of complementary feeding (i.e., focusing on intact fruits rather than fruits juices given their free sugars content) and recommended in the local dietary guidelines for infants under 2-years old [41]. Future research on this topic should include studying the relevance of this natural source of dietary sweetness in later preferences and health. Regarding dairy products, they are also widely recommended for infants and young children, but little focus is put on the sweetness of these products. An important proportion of milk and milk-based drinks are sweetened (beyond the natural sweetness intensity of lactose) in the Chilean market, according to local reports in 2015 and 2016, 32% of products were considered ‘high in sugars’ (i.e., total sugars > 6 g/100 mL) [42], and 50% contained at least one NNS [19]. The relevance of beverages as a source of sweetness density at both ages is particularly worrying, and this result is consistent with a report ranking Chile as the top consumer of sugary-sweetened beverages [3], the reported intake of ultra-processed beverages in a group of Chilean preschoolers [43], as well as other reports showing high rates of ‘high in sugars’ beverages or NNS containing beverages [19,42]. This data should be considered for planning dietary recommendations and other measures aiming to improve the diet of infants and young children.

The associations of sweetness density with the intake of both sugars and NNS, support the use of this methodology to quantify dietary sweetness. Moreover, the increase and tracking of sweetness density between 12 and 36 months also support the methodology, given it was an expected result based on previous reports. Yuan et al. [18] in France (2005–2009), reported an increase of 127% in diet sweetness between 3 and 12 months of age. These authors assigned sweetness intensity using the Martin et al. database (used as a secondary database in the present study), considering dietary information on the frequency of consumption. This increase in diet sweetness with age was also reported by Nguyen et al. [44], who described an increase in the proportion of energy coming from sweet foods from 20 to 29% between 12 and 24 months old in a sample of infants from the Netherlands (2003–2007). In this study, the van Langeveld et al. database was used (primary database for the current study). We have not identified follow-up studies of diet sweetness among older age children to compare our results.

In a cross-sectional study carried out in an adult French population, a positive association was described between sweet preference and excess weight [45]. In our study, we did not observe an association between sweetness density of diet (which could be a proxy for sweetness preference, although at this age-particularly at 12 months the diet is defined almost exclusively by parents) and BMI category. The difference between these results may be due to the ages considered, in addition to important methodological differences (e.g., sweetness preference evaluated by survey alone).

Despite the relevance currently given to diet sweetness, especially for the pediatric population, there are few methodologies to conduct an evaluation of diet sweetness (only 3 databases of food sweetness intensity could be identified), which results in few published studies on the subject. Thus, evidence is still limited regarding the impact of exposure to sweet taste, for example, on subsequent acceptance and preference for sweet foods and beverages [15]. Follow-up studies will be essential to study diet sweetness as an exposure and possible effects in the medium- and long-term.

Regarding the weaknesses of our study, it should be mentioned that a large part of the sweetness data came from secondary databases created in other countries. Thus, there could be differences due to product formulations. The challenge remains to advance in building national databases on the sweetness intensity of foods locally available. Another weakness was that the sweetness intensity evaluation for all foods was conducted by a trained panel of adults but applied to the diet of children. Young children, because of their cognitive and sensory abilities, cannot conduct this type of assessment [46]. Some studies imply that perceptions could be different between children, adolescents and adults [47]. On the other hand, the estimation of diet sweetness density considered energy intake as a way of adjusting for total intake; however, this might be a problem when the presence of NNS in foods/beverages is high (given they provide reduced or null energy to the total intake). We repeated the analyses using an alternative sweetness density indicator, which considered grams of foods/beverages (assuming a beverage density of 1 g/mL) as the denominator, obtaining similar results. Further studies could consider using this alternative indicator if it is more useful for their data and purpose. Finally, in our study, we only had one 24R per participant (one at each age), which may not represent the usual intake. However, 24R is considered the gold-standard method to evaluate diet [48]. Furthermore, we applied the multiple-step method, which allows for obtaining detailed information on the products consumed, both in quantities, as well as brands and varieties.

Among the strengths of the study, we highlight the use of a sensory panel that was subjected to selection tests and training, prior to the evaluation, which allowed for optimizing the internal consensus between evaluators and technique replication [49]. The use of methodologies similar to previously published work is also important, facilitating comparisons between studies. For example, we used sucrose solutions in the same concentrations as described in previous studies, from van Langeveld et al. and Martin et al., and used the same or analogous reference foods for the different training tests. We conducted training sessions for product sweetness intensity values and obtained similar results when comparing the values for some products between published databases and our evaluation by the trained panel. In addition, there was internal consistency in the methodology used since, as expected, diet sweetness density was associated with the consumption of total sugars and having consumed at least one NNS. This is the first quantification of diet sweetness in Chile and, as far as we know, one of the few studies evaluating diet sweetness over time in a sample of infants. 

In conclusion, we describe a methodology for assessing dietary sweetness, using an updated database with sweetness intensity values for foods and beverages. The application of this method in dietary data from 12- and 36-months old children showed that diet sweetness was associated with sugars and NNS consumption at both ages. Moreover, the indicator could be tracked over time and we observed that dietary sweetness increased with age. Our results validate the proposed method, in a context with no gold standard being described for assessing dietary sweetness. Future studies using this method in other dietary contexts and with explicit research questions could provide further validation. Monitoring diet sweetness during childhood is key to studying the impact that early exposure to high levels of sweetness may have on future health and diet outcomes.

## Figures and Tables

**Table 1 nutrients-14-01447-t001:** Comparison of sweetness intensity values of products reported in the primary database and evaluated by trained panel in the current study.

Product	Sweetness Intensity Value Reported in Primary Database	Sweetness Intensity Value Measured in Current Study
Strawberry jam	74	71
Orange juice	31	32
Whole milk	12	12

**Table 2 nutrients-14-01447-t002:** Examples of sweetness intensity values for select foods/liquids reported in the dietary recalls.

Food or Liquid	Sweetness Intensity Value
Condensed milk †	88
Jam †	74
Flavorings (e.g., chocolate powder) ¥	65
Carmel “Manjar” *	64
Soda (regular and diet) ¥	51
Ice cream ¥	46
Cola drinks ¥	44
Breakfast cereal ¥	41
Juice (regular and diet) ¥	40
Sweet purees *	38
Cookies ¥	35
Fruits ¥	30
Banana ¥	30
Yogurt ¥	30
Apple ¥	20
Milk ¥	12
Peanuts †	8
Beef †	4
Bread †	4
Natural pasta †	3
Cheese †	2
Oil †	1

Of these sweetness intensity values, 20 come from van Langeveld et al. database (of which 12 corresponds to mean values of a group of similar food items (¥) and 8 correspond to unique products (†)), and two come from the sensory evaluation carried out for this study (*).

**Table 3 nutrients-14-01447-t003:** Participant characterization at 12 and 36 months of age (*n* = 436).

	12 Months	36 Months	*p*-Value **
Female, %	49.3	49.3	-
Male, %	50.7	50.7	-
BMI z-score, mean ± SE	0.74 ± 0.89	0.82 ± 1.09	<0.01
Normal weight, *n* (%)	273 (62.6)	262 (60.1)	<0.01
Overweight, *n* (%)	137 (31.4)	129 (29.6)	
Obesity, *n* (%)	26 (6.0)	45 (10.3)	
Maternal education ≤ 12 years, *n* (%)	191 (43.8)	172 (39.4)	<0.01
Maternal education > 12 years, *n* (%)	245 (56.2)	264 (60.6)	
Maternal weight status: Normal weight, *n* (%)	112 (28.1)	73 (19.9)	<0.01
Maternal weight status: Overweight, *n* (%)	122 (30.7)	125 (34.1)	
Maternal weight status: Obese, *n* (%)	164 (41.2)	169 (46.0)	

BMI: body mass index. BMI z-score according to WHO standards. Weight status: Normal weight, BMI z-score < +1SD; Overweight, BMIz-score ≥ +1SD and<+2SD; Obesity, BMI z-score ≥ +2SD. *n* = 398 at 12 months; *n* = 367 at 36 months. Differences in sample sizes relates to the fact that participants may have been accompanied by another family member (e.g., father, grandmother). Maternal BMI category: Normal BMI, <24.9; Overweight, BMI = 24.9–29.9; Obesity, BMI > 30. ** McNemar test.

**Table 4 nutrients-14-01447-t004:** Association between total sugars, non-nutritive sweetener consumption and sweetness density.

Predictor Variable	Sweetness Density	
	12 Months	36 Months
Total sugars, % of calories	24.6 [17.3–31.9] *	44.7 [33.2–56.1] *
Consumed non-nutritive sweetener †	243.8 [107.4–380.2] *	715.5 [494.8–936.2] *

Values represents β coefficients [95% confidence interval]. Outcome variable: Sweetness density. * *p* < 0.01. † reference group = does not consume non-nutritive sweetener.

**Table 5 nutrients-14-01447-t005:** Primary food sources of diet sweetness at 12 and 36 months of age.

Food Groups	12 Months	36 Months
Fruits	27.3	13.7
Beverages	19.3	32.2
Vegetables/algae and mushrooms	17.7	5.7
Baby foods	13.3	2.7
Dairy and substitutes (e.g., soy/almond drink)	10.8	28.6
Sugars and candy	3.6	6.2
Grains and bread	3.2	6.0
Meat and substitutes (e.g., soy)	2.1	1.5

Values represent % of sweetness density coming from each food group, considering total sweetness density of the diet as denominator.

## Data Availability

Not applicable.

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
