# Peer review of "Sweetness of Chilean Infants’ Diets: Methodology and Description"

_nutrients, 2022, doi:10.3390/nu14071447_

Round 1
Reviewer 1 Report
The manuscript entitles: “Sweetness of Chilean infants diets: methodology and description” was submitted to Nutrients journal for review. The manuscript is an interesting piece of work, however I have the following comments which need to be addressed before publishing in Nutrients journal:
*introduction
Authors focused on health impact of sugar consumption but ignored the health impact of consuming NNS by children as some of artificial sweeteners have adverse effect/not advised for child consumption. Please add one to two lines about that fact.
*Study design:
1)Authors in this current work relied on already published work by van Langeveld et al., that created a data base for children and adults, in their study design. Authors used food products reported in van Langeveld et al. as shown in table 2 which are not suitable for 12 months babies. For example, how a 12 months old baby would eat/drink breakfast cereal, cola drinks and soda? this is a clear fault in the study design. Please revisit table 2 and from 5,649 products investigated add logical food products that 12 and 36 months infant would consume in addition of showing the sweetness densities.
2) 12 months babies are usually breastfeed (first three to six months) and/or given baby formula solely in the first months of their life and no actual food intake at this early stage of life (no teeth to chew). Authors did not mention this point in the study design and sample consideration (exclusion/inclusion criteria)?
3) authors did not explain why they chose these two age groups 12 and 36 months? I feel that these two groups cannot be treated the same in term of food intake. It was clearly mentioned in the manuscript in page 8 lines 310 and 3011 that: “although at this age the diet is defined almost exclusively by parents” and this is totally correct for 12 months infants. While the 36 months of age have more degree of independence in choosing and rejecting what they want to eat. So is it logic to compare between these specific age groups?.
* Results
1) tables 2 and 5 should have a proper link in between. In table 2, there must be another column related to “sweet density due to NNS containing products” besides the already existing reported sweet density due to sugar. This will give a proper understanding of table 5.
2) please indicate that the values in table 2 are “means”.
3) what is the point of table 4, I mean it is very obvious that 12 months old infant are naturally eating less food that 36 months old infants as I highlighted this point earlier above. I suggest to remove/replace this table and write two to three lines in as text.
4) table ligand for table 6 does not reflect what is presented in the table. Please rewrite the ligand to reflect the fact that the numbers are “the frequency of intake” of various food products by 12 months and 36 months infants.
5) why authors did not mention the type NNS identified in infants products under investigation? It would be useful for the reader to be notified in one to two text lines about the “most common type of NNS” found in such products.
6) page 6 lines 220 and 221 = 12 or 36 months – change = 12 and 36 months
Discussion:
Authors mentioned in page 9 line 347: “the first study to evaluate diet sweetness over time in a sample of infants” how this is indicated/reflected in the manuscript?
Reviewer 2 Report
Dear Authors,
The article presented for review touches upon an important issue concerning children's health, as well as contains interesting results.
In order to publish it, one of the following changes is required:
- it should be explained where the consumption of fruit and beverages is so high,
- why finally the group consisted of 436 children?
- is BMI for children correct? Isn't it more correct to use percentile grids?
- table 3: why are the numbers different than 100%? 99.9 or 100.1%?
- why is only the percentage of women given?
- was the McNemar test described in the statistical methods?
- BMI ranges are probably misspelled, as they range below 24.9 and then from 24.5,
- line 254 - why did the authors adopt p> 0.01 and not p> 0.05?
- there are no clear conclusions summarizing the study,
- the literature is not at all compliant with the journal's requirements.
Minor formatting suggestions:
- in the affiliation, at the e-mail addresses, the authors' initials should be entered,
- "n" or count - it should be written in italics,
- line 175 - do you need the word "formula"?
- there must be a space between the tables and this text, e.g. after table 1,
- line 215 - "three" should not be written with a number?
Round 2
Reviewer 1 Report
- Page 2 line 75: Moreover, the different dietary characteristics between 12 and 36 months of age provide a broader spectrum for applying the methodology. This statement is not clear … please explain “broader spectrum for applying the methodology”
- The authors response to reviewer’s comments in relation to type of NNS can be added as a paragraph to page 8 start of line 296.
Author Response
Thank you once again for your suggestions.
Point 1: Page 2 line 75: Moreover, the different dietary characteristics between 12 and 36 months of age provide a broader spectrum for applying the methodology. This statement is not clear … please explain “broader spectrum for applying the methodology”
Response: We have revised the complete statement for improving clarity (L75-77).
Point 2: The authors response to reviewer’s comments in relation to type of NNS can be added as a paragraph to page 8 start of line 296.
Response: We have added the information where suggested by the reviewer (L297-299)
Reviewer 2 Report
Dear Authors,
Thank you for considering my suggestions. Minor editorial errors should be corrected at the proof stage (eg "n" should be italicized everywhere).
Author Response
We have edited 'n' as suggested by the reviewer and will make any further edition that might be asked by the editorial office.